# Quantitative Determination of Diosmin in Tablets by Infrared and Raman Spectroscopy

**DOI:** 10.3390/molecules27238276

**Published:** 2022-11-27

**Authors:** Sonia Pielorz, Magdalena Węglińska, Sylwester Mazurek, Roman Szostak

**Affiliations:** Department of Chemistry, University of Wrocław, 14 F. Joliot-Curie, 50-383 Wrocław, Poland

**Keywords:** diosmin, Raman, infrared, NIR, quantitative analysis, pharmaceutical preparations, PCA, PLS

## Abstract

Diosmin is widely used in the treatment of chronic venous diseases and hemorrhoids. Based on Raman and infrared reflection spectra of powdered tablets in the mid- and near-infrared regions and results of reference high-performance liquid chromatographic analysis, partial least squares models that enable fast and reliable quantification of the studied active ingredient in tablets, without the need for extraction, were elaborated. Eight commercial preparations containing diosmin in the 66–92% (*w*/*w*) range were analyzed. In order to assess and compare the quality of the developed chemometric models, the relative standard errors of prediction for calibration and validation sets were calculated. We found these errors to be in the 1.0–2.4% range for the three spectroscopic techniques used. Diosmin content in the analyzed preparations was obtained with recoveries in the 99.5–100.5% range.

## 1. Introduction

Diosmin, a 7-rhamnoglucoside of 3′,5,7-trihydroxy-4′-methoxyflavone, IUPAC name of 5-hydroxy-2-(3-hydroxy-4-methoxyphenyl)-7-[(2S,3R,4S,5S,6R)-3,4,5-trihydroxy- 6-[[(2R,3R,4R,5R,6S)-3,4,5-trihydroxy-6-methyloxan-2-yl]oxymethyl]oxan-2-yl]oxychro-en-4-one, is a chemical compound belonging to a group of flavonoids originating in the Rutaceae family [1]. It was first isolated in 1925 from Scrophularia nodosa L. [2]. Diosmin is a gray-yellow powder. Similar to other flavonoids, it dissolves poorly in polar and nonpolar protic solvents but much better in aprotic solvents (e.g., DMSO) [3]. The poor solubility of diosmin in most solvents creates a problem when it is used as a drug. For this reason, various technological processes are applied, including micronization, to increase the solubility and thus bioavailability of this flavonoid [4]. Currently, diosmin is isolated from the flesh, skin and seeds of citrus fruits (Citrus genus), mainly from the fruits of bitter orange (Citrus aurantium). It is obtained synthetically from hesperidin through treatment with an aqueous solution of sodium hydroxide in the presence of iodine and pyridine with an efficiency of 66% or in the process of acetyl hesperidin bromination using N-bromosuccinimide, benzoyl peroxide and chloroform, achieving an efficiency equal to 44%, as well as using ionic liquids [5,6]. The obtained compound may be accompanied by various impurities. Their presence affects the quality of the final products. Depending on the method used to process plant material and the synthetic route, diosmin may be contaminated by diosmetin, rhamnose, glucose, eriocitrin and other compounds [7,8]. It is worth noting that diosmin is usually accompanied by hesperidin, from which it is synthesized.

Diosmin is widely used in the treatment of chronic venous insufficiency, hemorrhoids, lymphoedema and varicose veins [9]. Preparations containing diosmin prevent inflammation [10,11] and intensify lymphatic drainage supporting microcirculation [12,13]. These actions reduce symptoms such as swelling, heaviness, cramps and pain in the calves, accelerate the healing of venous ulcers and improve quality of life [14,15]. Diosmin also demonstrates antioxidant [16], antiproliferative [17] and antidiabetic [18] properties. Hundreds of pharmaceutical preparations containing from 300 to 1000 mg of this active substance in one tablet are distributed all over the world. Therefore, the existing analytical methods are constantly being improved, and new methods are sought for the analysis of this active compound. Several methods can be applied for the determination of diosmin in pharmaceutical preparations. The most frequently used is high-performance liquid chromatography [1]. Among other techniques, thin-layer chromatography [19], voltammetry [20,21] and different spectroscopic methods can be listed [3,22,23]. Prior to analysis, using the enumerated methods, diosmin has to be extracted from the analyzed drug. Extraction is not required when vibrational spectroscopy is applied. Fourier transform infrared and Raman spectroscopy, assisted by multidimensional data analysis techniques, become more and more widely used in the pharmaceutical industry, both for qualitative and quantitative analysis of interesting compounds. These methods are easy to perform and time-efficient compared to chromatographic methods. They enable the analysis of active substances present in pharmaceuticals without active substance extraction or additional sample pre-treatment, which significantly shortens and simplifies the analysis [24,25,26].

Although the application of near-infrared spectroscopy (NIR) is well established in many areas [27,28,29], other, less common techniques, including attenuated total reflection (ATR) and diffuse reflectance infrared Fourier transform spectroscopy (DRIFTS) in the mid-infrared region (MIR) and Raman spectroscopy, offer unique possibilities with regard to medicines, foods, body fluids and plant and animal tissue analysis [25,26,29,30,31,32,33]. The most important advantages of these methods include minimal requirements for sample preparation, simplicity of implementation, short analysis time and the ability to automate the process. In combination with chemometrics, they enable fast and detailed qualitative and quantitative analysis of a variety of objects, often in their native form [33,34,35].

The stable crystal form of diosmin is its monohydrate (Figure 1). The anhydrous form can be prepared by heating DSNM at 110–140 °C. The obtained form is hygroscopic and transforms on air into a monohydrate within 72 h [36]. As a result, diosmin monohydrate is an active pharmaceutical ingredient (API) in commercial preparations.

## 2. Experiment

### 2.1. Materials and Sample Preparation

Diosmin was isolated from Preparation 1 (Appendix A). According to manufacturer’s declaration, except for diosmin consisting of more than 89% of the tablet weight, magnesium stearate, polyvinyl alcohol and sodium croscarmellose were present. After weighing, tablets were thoroughly pulverized using an FW100 grinder (ChemLand, Stargard, Poland). Next, 14 mL of 0.5 M sodium hydroxide solution was added per tablet. Then, the mixture was vigorously stirred for 15 min, and it was allowed to sit until the next day. The solution was gravity filtered. Next, twice as much demineralized water was added. After 15 min of stirring, 1.2 M hydrochloric acid was added to the solution, to a pH value of 7, to precipitate the diosmin. The pH of the solution was controlled with universal papers. The mixture was filtered through a Buchner funnel. The resulting precipitate was washed with demineralized water and dried in a desiccator at room temperature. Its purity was checked using the HPLC method. The diosmin was isolated in two series with an efficiency reaching 90%. Raman, MIR and NIR spectra of isolated diosmin were identical with the respective spectra of the diosmin analytical standard (Sigma-Aldrich, Saint Louis, USA). A set of 83 calibration samples, consisting of diosmin, magnesium stearate (Sigma-Aldrich, Saint Louis, MO, USA), polyvinyl alcohol (POCH, Gliwice, Poland) and croscarmellose sodium (Sigma-Aldrich, Saint Louis, MO, USA) (Appendix A) was prepared. Raman, MIR and NIR spectra of diosmin and the remaining ingredients are shown in Appendix A.

### 2.2. Reference Analysis

The content of the diosmin and hesperidin was determined (Appendix A) in the analyzed pharmaceutical preparations using high-performance liquid chromatography with diode-array detection (HPLC-DAD). The appropriate amount of powdered tablet, for which the content of diosmin was about 450 mg, was dissolved in 50 mL of a 0.5 M NaOH solution, in a 250 mL volumetric flask. Then, the solution was sonicated for 15 min. After the sample reached room temperature, the volumetric flask was refilled with a mixture of 0.01 M trisodium buffer (pH = 12.4) and methanol (60:40, *v*/*v*). A total of 25 mL of the prepared solution was diluted again to 100 mL with the aforementioned solvent mixture. The determination of diosmin and hesperidin concentration was carried out in the analyzed preparations with the X-Bridge RP C18 3.5 µm column, 150 × 4.6 mm. The injection volume of the sample was 30 µL. For the separation of active ingredients, the mixture of methanol: water (+0.1% H_3_PO_4_, 50:50 *v*/*v*) was used as the mobile phase. The mobile phases were passed through a 3.5 µm thick membrane filter, and the flow rate was adjusted to 0.7 mL/min. The active substance concentration was determined by measuring the absorbance at 270 nm [37,38].

### 2.3. Apparatus

A Nicolet Magna 860 FT-IR spectrometer with a Nicolet Raman unit (Thermo Nicolet, Madison, WI, USA) was used to register the spectra. DRIFTS reflection spectra were obtained using DTGS detector and a Seagull (Harrick, New York, NY, USA) optical assembly set to DRIFTS mode. A KBr beamsplitter was applied to measure the mid-infrared spectra, and a CaF_2_ beamsplitter in the NIR region was used. A total of 128 interferograms for NIR measurements and 256 for MIR and Raman were averaged. Interferograms were Happ–Genzel apodized and Fourier transformed using a zero filling factor of 2, giving spectra in the ranges of 400–4000 cm^−1^ for MIR, 3700–10,000 cm^−1^ for NIR and 100–3700 cm^−1^ for Raman with a resolution of 4 cm^−1^. To register the Raman spectra, an InGaAs detector, CaF_2_ beamsplitter, 180° backscattering geometry and a rotating sample holder enabling sample rotation at a constant speed of 200 rpm were applied. The spectra were excited using an Nd: YVO_4_ laser with a power of about 400 mW at the sample. All measurements were performed at room temperature. 

For the DRIFTS measurements, samples were diluted with potassium bromide, calcined for two hours at 200 °C, at a ratio of 1:49. Pellets for the Raman spectra measurement were prepared.

HPLC analyses were performed using Waters HPLC 600 Quat Pump, 717 Plus Autosampler and 2996 Detector chromatograph (Markham, ON, Canada).

### 2.4. Software and Numerical Data Treatment

The principal components analysis (PCA) of the obtained data was performed using the PLS Toolbox (ver. 6.2, Eigenvector Research, Wenatchee, WA, USA) in a Matlab R2010a environment (MathWorks, Natwick, MA, USA). TQ Analyst (ver.7, Nicolet, Madison, WI, USA) chemometrics software was used to construct the partial least squares (PLS) models [39,40].

In order to characterize and compare the prognostic abilities of the developed calibration models, the relative standard error of prediction (RSEP) was calculated for the calibration and validation samples, according to the equation:(1)$$\mathrm{RSEP}\left(\%\right)=\sqrt{\frac{\sum_{i=1}^{n}{\left(C_{i}-C_{i}^{A}\right)}^{2}}{\sum_{i=1}^{n}C_{i}^{A^{2}}}}\times 100,$$
where C_i_^A^ is the actual content of the active substance in the preparation, C_i_ is content determined on the basis of the model, and *n* is the number of samples [26].

Density functional theory (DFT) calculations were performed at the B3LYP/6-311++G(d,p) level of theory using the Gaussian09 suite of programs [41]. We found all stationary points to be true minima because no imaginary frequencies were obtained.

## 3. Results and Discussion

### 3.1. Vibrational Spectra

Raman spectra recorded for the hydrated and anhydrous forms of this flavonoid (Appendix A) do not show significant changes in the position, shape and intensity of the main vibrational bands. More pronounced differences are observed in the IR spectra (Appendix A), mainly in the 865–1085, 1350–1670 and 2800–3700 cm^−1^ wavenumber ranges, corresponding to ν(C-O), ν(C-OH) stretching, δ(C-OH), δ(CC-H) deformation and ν(O-H) stretching vibrations [42,43]. 

These changes result from the presence of a water molecule in the structure of the crystalline form of hydrated diosmin [44]. The most intense bands located in the Raman spectrum at 1501, 1572 and 1611 cm^−1^ and at 1514, 1567 and 1611 cm^−1^ in the DRIFTS/MIR spectrum are attributed to the ν(C=C) stretching of the phenolic ring and δ(C-OH) and δ(CC-H) deformation vibrations. Vibrations of the carbonyl group ν(C=O) result in a band with a maximum at 1660 cm^−1^. Bands with a maximum at 1142 and 1289 cm^−1^ in the Raman spectrum and at 1142 cm^−1^ in the MIR spectrum are related to the vibrational movements of the C-O-C and C-OH fragments present in diosmin molecules. Stretching vibrations of hydroxyl groups ν(O-H) give bands located in the 3200–3700 cm^−1^ range. Spectral features related to vibrations of the C-OH fragments are observed at 1010, 1074 and 1098 cm^−1^ in the MIR spectrum. Bands at 1453 and 1470 cm^−1^ are assigned to δ(CH_3_) vibrations of the methyl groups and those located in the 2850–3080 cm^−1^ range in Raman and MIR spectra to ν(C-H) stretching vibrations [42,43]. The assignment of the most important diosmin vibrational bands is summarized in Appendix A. 

The theoretical and experimental IR and Raman spectra of the hydrated (DSNM) and anhydrous (DSNA) forms of diosmin (Figure 2 and Appendix A) are very similar, regarding the number and position of the bands, except for the ν(O-H) stretching vibration region. In the theoretical DSNM Raman spectrum, a weak band, located at 1079 cm^−1^, is shifted toward lower wavenumbers relative to the corresponding DSNA band. This band reflects movements of the C-OH fragments of the compound (Appendix A). Some changes in the relative intensity of bands are observed, mainly in the 1060–1150, 1530–1670 and 2840–2970 cm^−1^ wavenumber regions. A similar difference is observed in the theoretical IR spectra of the hydrated and anhydrous forms of diosmin (Appendix A). In the DSNM spectra, an additional band of medium intensity, with a maximum at 1594 cm^−1^ appears, not observed in the spectrum of its anhydrous form. This band is attributed to deformation vibrations of the hydroxyl group δ(H-OH) of the water molecule present in DSNM. Bands of the symmetrical and asymmetrical stretching vibrations of the water molecule with a maximum at 3507, 3587 and 3812 cm^−1^ are present. The band located in the 1015–1045 cm^−1^ range in the DSNA spectrum consists of two bands, while in the DSNM spectrum, a single, symmetrical band is observed. The most pronounced differences in the relative intensities of the bands are observed in the 965–1120, 1360–1430, 1585–1610 and 2800–3700 cm^−1^ ranges. 

Despite the superficial similarity of IR spectra of both diosmin forms (Appendix A), the contribution of the water molecule present in the crystal structure of DSNM is clearly visible in the difference spectrum. A broad band located approximately at 3300 cm^−1^ can be assigned to the ν(O–H) stretching vibrations of the hydrogen-bonded water molecule. Additionally, in the difference spectrum, the contribution resulting from changes in a δ(CC–H) deformation and ν(C=C) stretching vibrations of the hydrated and anhydrous form carbohydrate fragment can be noticed in the 1375–1550 cm^−1^ wavenumber range. In the IR difference spectrum obtained from the DFT calculation, bands with maxima at 1594, 3507, 3587 and 3812 cm^−1^ are observed.

The recorded NIR spectra of the hydrated and anhydrous forms of diosmin (Appendix A) show no pronounced differences, except in the 4400–5200 and 6200–7000 cm^−1^ ranges. A broad band of medium intensity, with a maximum at 6737 cm^−1^, can be attributed to the first overtone of the hydroxyl group stretching vibrations and those located at around 6000 cm^−1^ to the first overtones of the ν(C–H) vibrations. In the 4800–4890 cm^−1^ frequency range, combination bands of ν(O–H) and ν(C=C) stretching vibrations are present [44,45,46,47]. In the NIR difference spectrum of the hydrated and anhydrous forms of diosmin, bands that can be attributed to the vibrations of the water molecule bonded to the diosmin carbohydrate fragment are observed at 4490 and 6950 cm^−1^ [48].

FT-Raman, DRIFTS/MIR and DRIFTS/NIR spectra of the representative calibration sample and the studied pharmaceutical preparations are shown in Figure 3. The analyzed preparations, apart from API, the content of which varied in the 65.3–89.4% range as declared by the manufacturers, contained croscarmellose sodium, carboxymethyl starch, microcrystalline cellulose or starch, polyvinyl alcohol or povidone and magnesium stearate or stearic acid. In some of them, colloidal silica and talk were also present. Detailed data on the composition of the analyzed preparations is presented in Appendix A. Despite the slightly different chemical compositions of the analyzed tablets, their vibrational spectra are similar to each other. These differences become visible in the space of the first two principal components (PC1/PC2) (Figure 3). Appendix A shows the loadings plots for PC1 and PC1 principal components obtained on the basis of Raman, MIR and NIR spectra of calibration samples. 

### 3.2. Chemometric Analysis

Keeping in mind that diosmin contributes from approximately 65 to 90% of the analyzed preparations and the fact that the excipients present are the same or related chemical compounds, we have decided to prepare calibration samples composed of diosmin, polyvinyl alcohol, sodium croscarmellose and magnesium stearate. Their mass fractions ranged from 0.603 to 0.926 for diosmin, from 0.005 to 0.054 for magnesium stearate, from 0.016 to 0.222 for polyvinyl alcohol and from 0.017 to 0.202 for sodium croscarmellose. In the prepared mixtures, concentrations of individual components did not correlate with each other. A set of 83 samples was used to construct PLS models, with approximately 25% of them selected using a bootstrap method and a sample distribution in PC1/PC2 space as test samples. Some of them have been omitted, but not more than 10 in each case. We summarize the type of spectra preprocessing and spectral ranges selected for analysis in Table 1 and Appendix A. Appendix A shows regression vectors obtained on the basis of Raman, MIR and NIR spectra of calibration samples.

To assess the spectral identity of the prepared calibration mixtures and the commercial diosmin preparations, PCA was performed. For each of the applied spectroscopic techniques, points corresponding to the analyzed preparations, as shown in Figure 3, are evenly distributed in the PC1/PC2 space, constructed based on calibration sample spectra. To determine diosmin content, PLS models were developed for each of the analyzed preparations and techniques applied separately. Spectral ranges selected for the analysis differ slightly depending on the preparation, due to the differences in the composition of tablets originating from different manufacturers, as we have mentioned before. Elaborated PLS models are of similar quality within a given spectroscopic technique. Relative standard errors of prediction are in the range of 1.0–2.4% and 1.1–2.4% for the calibration and test sample sets, respectively (Table 1 and Appendix A). Internal validation of the models using a leave-one-out (LOO) procedure resulted in correlation coefficients of cross-validation in the 0.944–0.964 range. Characteristics of the constructed models are presented in Figure 4, Table 1 and Appendix A. The number of latent variables (5, 6 or 7) was selected based on the RMSECV plots. We determined the diosmin content on the basis of vibrational spectra of the analyzed tablets with a recovery of 99.5–100.5% (Figure 5) with a standard deviation varying in the 0.6–3.4% range (Appendix A). Selectivity ratio (SR), presented in Figure 4, shows spectral contributions of the variables in the projection used in the PLS models. For Raman data, the highest contributions are observed in the frequency range of 1000–1750 cm^−1^ and 490–790 cm^−1^. For MIR data, regions 1000–1800 cm^−1^ and 3000–3700 cm^−1^ are the most important, while for NIR data, these contributions are large in the 4050–5300 and 6900–7250 cm^−1^ ranges.

A similar analysis was performed based on ATR data for selected diosmin preparations, and results of comparable quality were obtained.

## 4. Conclusions

Here, for the first time, the suitability of Raman and NIR spectroscopy for the determination of diosmin content in intact pharmaceutical preparations was demonstrated. Eight commercial tablets containing diosmin as an active ingredient in the 66–92% (*w*/*w*) range were successfully quantified using PLS models based on Raman and infrared reflection spectra of powdered tablets in the mid- and near-infrared regions, with an error below 2.4%. The concentration of diosmin in commercial pharmaceutical preparations determined based on PLS models is consistent with the results of the reference analysis with a recovery of 99.5–100.5%. The quality of determinations is comparable for the three methods used. The described procedure enables efficient, fast and reliable quantification of active ingredients in tablets, supporting the analysis of pharmaceutical products containing diosmin.

## Figures and Tables

**Figure 1 molecules-27-08276-f001:**
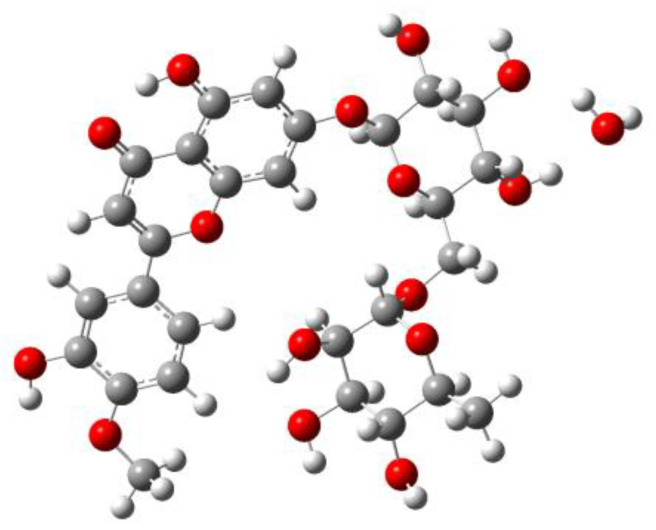
Structure of diosmin hydrate (DSNM).

**Figure 2 molecules-27-08276-f002:**
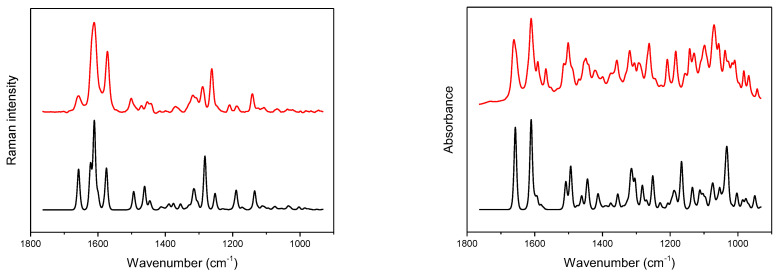
Experimental (red) and calculated (black) Raman (**left**) and IR (**right**) spectra of hydrated diosmin (DSNM); for calculated spectra abscissa scale multiplied by a factor of 0.98; Gauss-Lorentz profile with a half-width of 8.9 cm^−1^ was used to obtain calculated spectra.

**Figure 3 molecules-27-08276-f003:**
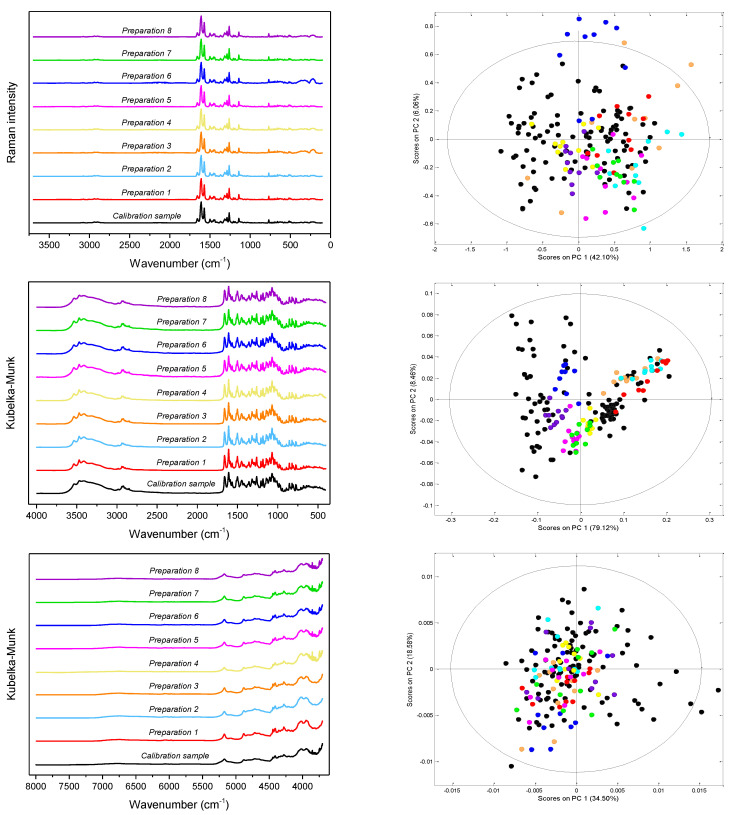
From the top: Raman, MIR and NIR spectra (**left**) of the studied preparations and distribution of samples in the PC1/PC2 space with the 99% confidence interval (**right**); black dots—calibration mixtures, colored dots—pharmaceutical tablets.

**Figure 4 molecules-27-08276-f004:**
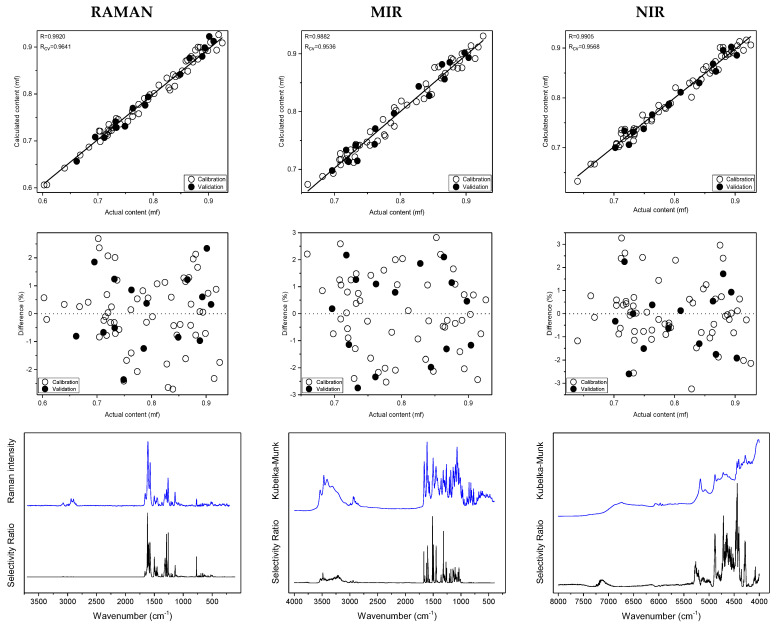
Prediction plots, relative errors, experimental spectra of hydrated diosmin (DSNM) (blue) and selectivity ratio (SR) (black) obtained on the basis of Raman, MIR and NIR spectra of calibration samples for Preparation 1; from the top.

**Figure 5 molecules-27-08276-f005:**
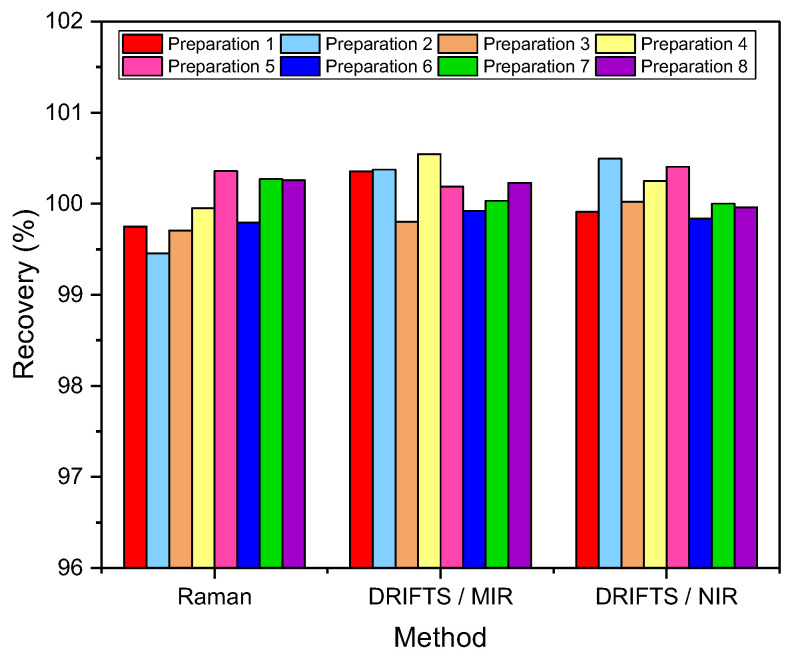
Diosmin recovery (*n* = 10).

**Table 1 molecules-27-08276-t001:** Parameters of PLS models developed for Preparation 1.

Parameter	RAMAN	MIR	NIR
R_cal_	0.9920	0.9882	0.9905
R_test_	0.9932	0.9838	0.9877
R_cv_	0.9641	0.9536	0.9568
RSEP_cal_	1.30	1.41	1.32
RSEP_test_	1.25	1.59	1.39
Number of PLS factors	5	7	7
Wavenumber range [cm^−1^]	488–963	1478–1633	3811–3957
	3036–3119	2391–3452	4379–4762
			6153–6683
Normalization	SNV	None	SNV

R—correlation coefficient, R_CV_—correlation coefficient of cross-validation, cal—calibration set, test—test set, SNV—standard normal variate.

## Data Availability

Not applicable.

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
