# Peer review of "Quantitative Determination of Diosmin in Tablets by Infrared and Raman Spectroscopy"

_molecules, 2022, doi:10.3390/molecules27238276_

Round 1

Reviewer 1 Report

The authors of this manuscript used mid-infrared, near-infrared, and Raman spectroscopy to quantify the content of eight commercial diosmin tablets, The method is feasible and the results are reliable. However, there are two points that I think could be improved.

1, from the results, the linearity and error of mid-infrared is slightly greater than the other two methods, may I ask the authors whether they have tried the ATR method? Line 260 mentions the ATR method, but why is it not explained in more detail? From lines 123 and 124, it can be seen that the sample pre-treatment time is very long in order to do DRIFTS analysis. If the ATR method is used, it should be possible to analyze the sample directly and save time. The results should also be more satisfactory.

2. Are lines 263-266 giving the band range used to perform the PLS calculation? If so, it is recommended to optimize it again. The high content of diosmin in the tablets should give a better linearity than the current results.

Author Response

Ad. 1. We had no possibility to complete ATR measurements because our spectrometer had crashed.

Actually we have managed to perform approximately a half of the experiment for the first 3 preparations.

Ad. 2. As we have written 'diosmin contributes from approximately 65 to 90% of the analyzed preparations and the excipients present are the same or related chemical compounds' so we have decided to prepare calibration samples composed of only 4 compounds, namely diosmin, polyvinyl alcohol, sodium croscarmellose and magnesium stearate. Our calibration mixtures do not contain all excipients present in the quantified preparations. This is the reason why parameters of the elaborated PLS models are slightly worse than one can expect and why selection of spectral regions was so important.

Reviewer 2 Report

This article focuses on the application of several vibrational spectroscopic techniques for the quantification of diosmin in pharmaceutical solid dosage forms. While the topic is timely and interesting, and the manuscript is well written, there are number of points that need to be addressed before publishing in any journal:

a) With majority of focus allocated to the drug compound, introduction lacks information about similar applications of vibrational spectroscopic techniques and approaches for such analysis. This would also help better rationalize their choice for the study.

b) While it was generally easy to read the manuscript, I find the use of active voice in the experimental section a bit distracting, and I would recommend for authors to convert it. Additionally, all of the abbreviations should be introduced to the reader, for example, DRIFTS etc. I would also suggest to update the images with highlighting the major spectral peaks that are discussed and later utilized for the chemometric analysis as it would greatly improve their impact.

c) PCA could be expanded by also including the loadings plots that would simply make the science clearer. I understand that it is here used to just assess the compatibility between reference samples and analytes as well as select the validation samples, but it is always a good practice to complement scores plots in such a way.

d) While the PLS analysis is expertly done, I feel like that for readers less familiar with this type of analysis, the simultaneous use of terms “cross validation” and “validation” could cause some confusion, and, perhaps, validation sample set should be better named as test set. Additionally, is there a particular reason why the correlation coefficient characteristics for the predicted (test set) models are omitted. At the same time authors present RSEPcal, which in my mind doesn’t hold much practical value. Additionally, for selectivity ratio plots, are the features fully correlating with diosmin? Would it be useful to also present alternative representations such as regression coefficients, perhaps, in a smaller spectral window with highlighted bands that would correspond to ones, for example, discussed for Figure 2. Otherwise, the broad spectral scale and generally small size of the figures make it hard to fully interpret the analysis.  

e) The information about the commercial tablets could, perhaps, be improved. While I understand that it might not be possible to fully disclose their origin, the actual contents of other components could be denoted if known.

f) I feel that the substantial data sets procured by the authors could be further utilized by also exploring data fusion approaches (at least low-level fusion), considering the number of spectroscopic techniques used. This could significantly elevate the scientific impact of the manuscript and allow to assess potential advantages (or disadvantages) for using combinatory approaches.

g) The destructive aspect of the presented approach implies some serious limitations for real world applications in pharmaceutical industry. It would be useful for authors to expand the discussion, especially considering that, for example, transmission Raman spectroscopy has also been highlighted in the literature to hold great potential for such analysis without the aforementioned drawback (this also relates to point (a), where the introduction lacks information about previous studies and current state-of-art).

Author Response

Ad. a. We have added a short paragraph about applications of vibrational spectroscopy.

Ad. b. As suggested, we have transformed text to passive voice, we have explained abbreviations and we have marked the most important bands (Fig. S2).

Ad. c. We have added loadings plots (Fig. S4).

Ad. d. We have replaced expression 'validation set' by the suggested expression 'test set', although privately we prefer to use the first one. Correlation coefficients for the test have been added. To show better relation between spectra and selectivity ratio plots Fig. 4 has been modified. As suggested, regression vectors have been plotted (Fig. S5).

Ad. e. We do not know the exact composition of the commercial preparations, except for the amount of diosmin and a list of excipients present.

Ad. f. We agree that data fusion can help to improve quality of diosmin quantification, but in practice, it is much easier to use one spectrometer or to perform one experiment. As we have mentioned 'the quality of determinations is comparable for the three methods used', and quantification errors are rather low.

Ad. g. For plane tablets in the case of Raman and NIR experiment the only operation needed is milling. For coated ones it is necessary to remove coating first. Indeed, at least sometimes, transmission Raman experiment can be easier to perform.

Round 2

Reviewer 1 Report

The manuscript can be accepted in the revised version. However, I strongly recommend the authors to find an instrument and do the ATR analysis.

Author Response

As we have written 'A similar analysis was performed based on ATR data for selected diosmin preparations and results of comparable quality were obtained'. Namely, 'working versions' of elaborated models, for the 3 analyzed preparations, were characterized by Rcal, Rtest and RCV parameters in the 0.97-0.99, 0.94-0.95 and 0.93-0.95 ranges, respectively. RSEP errors for calibration and test sets were found to be below 1%. We hope that, in future, it will be possible to return to the analysis of this system, but probably in another context.

Reviewer 2 Report

The authors have done a great job implementing the changes, well done!

Author Response

OK